# The integration of communicable and non-communicable disease (CD-NCD) health services in Africa: A scoping review

Takhona G. Hlatshwako[1,2]*, Oluwabusolami Ale[1], John Chapola[1], Lirui Jiao[1], Shuyi Song[1], Audrey Yao[3], Leah Frerichs[1], Tara Templin[1,2]

**1** Department of Health Policy and Management, Gillings School of Global Public Health, University of North Carolina at Chapel Hill, Chapel Hill, North Carolina, United States of America, **2** Carolina Population Center, University of North Carolina at Chapel Hill, Chapel Hill, North Carolina, United States of America, **3** Department of Maternal and Child Health, Gillings School of Global Public Health, University of North Carolina at Chapel Hill, Chapel Hill, North Carolina, United States of America

\* takhona@live.unc.edu

## Abstract

Amid growing challenges in sustaining health programs in low-and-middle-income countries, and the growing burden of multimorbidity, there have been proposals to consolidate services for communicable and non-communicable diseases through integrated care models. However, there is still limited evidence on the intervention components, barriers, and facilitators of integrated care. This study aimed to review the literature on integrated care models implemented in African health systems over the last two decades, with the goal of informing future development in this area. A scoping review was conducted using the Arksey and O'Malley framework. We searched PubMed, MEDLINE, Cochrane Library, SCOPUS, and PsycINFO databases on June 6, 2025. Only peer-reviewed studies on integrated care models for people with at least one communicable disease and non-communicable disease in one or more African countries were considered. A codebook was developed to extract key information, and each study was coded by two independent reviewers. The Consolidated Framework for Implementation Research 2.0 was used to code the barriers and facilitators of integrated care models. The search yielded 419 publications between 2000 and 2025, of which 26 studies were included in this review. The most common integrated care model was the HIV-Hypertension-Diabetes model. Experimental evidence suggested mostly positive outcomes, such as improved blood pressure control and lower facility costs per patient. Key intervention components included training healthcare workers on integration and improving data collection and information systems. Developing NCD care guidelines was the most common facilitator for enhancing integration. The key barriers included limited staff and a lack of equipment, medicines, and diagnostics. Integrated care models are an increasingly used strategy for expanding access to health services in Africa; however, there are several challenges to their implementation. More intervention support and rigorous,

**Data availability statement:** The data was compiled from published studies in publicly available databases. The protocol was registered in OSF (https://doi.org/10.17605/OSF.IO/ZMR5K). All necessary components to replicate our analysis can be found at the OSF repository and within the manuscript and supplementary files (Search Strategy).

**Funding:** TH received support from the Royster Society of Fellows. OA, JC, LJ, SS, AY, and LF received no specific funding for this work. TT received support from 5R01HD101453-05 and P2CHD050924. The funders had no role in study design, data collection and analysis, decision to publish, or preparation of the manuscript.

**Competing interests:** The authors have declared that no competing interests exist.

community-engaged evaluations are needed to ensure their viability and sustainability in low-resourced settings.

## Introduction

Amid cuts to global health funding, particularly across many low-and middle-income countries (LMICs) in Africa, there is an urgent need to re-strategize and mobilize available resources to keep health programs running [1,2]. One proposed strategy is to consolidate health services for both communicable and non-communicable diseases (NCDs) through integrated care models [3,4]. Organizations such as the World Health Organization have made strong calls for people-centered integrated care in response to growing health challenges [5]. While many health systems in Africa have historically focused on communicable diseases such as HIV and TB [6], there is a growing burden of NCDs such as diabetes and heart disease [7], which must be urgently addressed to prevent mortality and morbidity.

Concerningly, the response to NCDs across many African countries remains largely fragmented [6]. Although the rise in NCDs is a global concern, over 75% of deaths due to NCDs occur in LMICs, many in Africa [8]. Most health systems in low-resource settings are underprepared for the growing burden of NCDs [9]. For example, while it is relatively easy to obtain an HIV test or TB treatment in many parts of Africa, accessing cancer screening is significantly more challenging [10]. In contrast, high-income countries like the United Kingdom have made substantial progress in providing care for NCDs through expansion of services offered at public clinics, often through integrated care initiatives [11]. However, advancements in expanding health services in low-resourced settings, especially in Africa, have been slower [12].

Notably, the co-occurrence of chronic diseases is becoming more common globally, otherwise known as multimorbidity [13]. Multimorbidity is one of the most complex problems in global health today, as the ageing population grows and the likelihood of living with more than one chronic condition also grows [14]. For instance, a cohort study by Chang and colleagues found that there was a high burden of NCDs among older people living with HIV in West and East Africa, with at least 50% having a second medical condition [15]. Evidently, there is an increased need for integrated care approaches to tackle the double burden of communicable and non-communicable diseases [16,17].

Integrated care models are health care innovations that aim to optimize the treatment and management of various health conditions, often at a single point of care. These models may hold promise for expanding care as the prevalence of NCDs increases, although more research is needed [18]. Integrated care models vary widely, and *integration* can be conceptualized in various ways, including as an innovation in health service delivery, organization, financing, and other health system functions [19]. In particular, integrated care models for health service delivery — the focus of this review— are often characterized by several key features, including coordination and collaboration among healthcare providers, communication across different healthcare levels, and a patient-centered focus to ensure overall well-being

[20]. Four countries in East Africa have already developed action plans calling for the integration of HIV and NCDs [21]. However, evidence on integrated care models is still limited, and tends to focus on the integration of HIV and NCDs. For instance, a recent scoping review synthesized evidence on intervention outcomes of integrated care for people living with HIV and NCDs, particularly in primary care settings [18]. However, there remains an evidence gap on the intervention components of integrated care models, structural and process outcomes, and opportunities for improvement across the primary to tertiary care continuum, particularly in low-resource settings [22].

This review aims to systematically map and synthesize experimental and observational evidence on integrated care interventions in Africa, which is still an emerging area of research in global health. Specifically, the review seeks to answer the research question: where and how have communicable disease (CD) services, such as HIV and TB services, been integrated with non-communicable disease (NCD) services in Africa? The review aimed to accomplish two objectives: 1) to identify published studies of integrated care in Africa, describing the intervention components, structural and process outcomes, and use of digital technology, and 2) to examine the barriers and facilitators to integration within African settings.

This review builds on previous work done by Chireshe [18] and Ojo [23] to further identify healthcare delivery models that integrate communicable disease services, including but not limited to HIV/AIDS services, and considers how these have been integrated with a broad range of NCD services and mental health care in the African region. By taking a broader scope than previous reviews, this review may provide additional insights for developing alternative health service delivery models in this region, as well as other settings.

## Methods

The methodology for the scoping review was informed by the Arksey and O'Malley framework. We also followed the PRISMA checklist for scoping reviews (PRISMA-ScR) [24,25]. In this review, *Integrated CD-NCD care* was defined as the combined provision of health services for both communicable and non-communicable diseases at a single point of delivery. We conceptualized the alternative as a narrow-focused program providing services for a single disease, such as an HIV clinic. A protocol for review was registered on Open Science Framework (https://doi.org/10.17605/OSF.IO/ZMR5K) [26].

### Literature search

The search strategy for this topic focused on four key concepts: communicable diseases, non-communicable diseases, integrated care, and the African region (North and Sub-Saharan Africa). Key search terms for each concept were identified (S1 Appendix). We used these terms to search PubMed on February 22, 2025 (documented in the protocol), and then replicated the search on PubMed, MEDLINE, Cochrane Library, SCOPUS, and PsycINFO databases on June 6, 2025 (final cut-off date). Boolean operators such as "AND" and "OR" were used to combine concepts and search terms, respectively.

### Selection of studies

Only peer-reviewed, original research studies for which the full text was available in English were considered (This language restriction did not exclude any papers). Both experimental and observational studies were considered. Studies published since 2000 were considered. Studies published before 2000 were excluded as they were considered to have little relevance for the research question.

*Summary of eligibility criteria:*

1. Peer-reviewed, original research studies on an integrated care model (receiving care for more than one condition at a single point of delivery) published between 2000 and 2025

2. Focused on at least one communicable disease and one non-communicable disease, including mental health conditions.

3. Focused on one or more countries in the African region

4. Published in English

Abstracts were screened for eligibility, and eligible studies were exported to a reference manager for full-text review. References of eligible studies were also searched for additional relevant papers.

## Synthesizing data

Data extraction was conducted by two independent reviewers, who each extracted the following data from each study: author, year of publication, country, study design, disease areas, number of total participants, intervention components, process outcomes, barriers, facilitators, and the use of digital technology (which we operationalized as both an intervention component and facilitator). The reviewers used the codebook described in the review protocol [26].

To develop the codebook, we defined key components of an integrated care model using a Delphi consensus study by McCombe et al. [20]. These components include improved data collection and surveillance, strong drug procurement systems, availability of equipment and tests, community-based education programs, training of health care workers, and task-shifting, among others. McCombe and colleagues' list was used to examine the studies of integrated care included in the review for these intervention components [20]. Other notable structural and process outcomes that were extracted include: treatment adherence, patient engagement and retention in care, staff training, timeliness of care, and costs. The coding of barriers and facilitators to implementation was guided by the Consolidated Framework for Implementation Research (CFIR) 2.0.[27] The CFIR framework is a widely used implementation science tool that helps to characterize the key features of interventions to inform future implementation and evaluation.

All the data was synthesized into an Excel spreadsheet. We took an inclusive approach to discrepancies in reviewer assessments, and conflicts were resolved by a third reviewer (TH). A large language model (OpenAI's ChatGPT GPT-4) was used for text categorization and frequency counting, particularly to generate counts for the number of times each code appeared in the data. The prompts and output of the LLM model are shown in S2 Appendix. The output of the LLM model was double-checked by hand to ensure accuracy. Quality assessment of experimental studies was conducted using the Critical Appraisal Skills Programme (CASP) checklist [28] to assess quality and risk of bias.

As this review utilized published peer-reviewed studies, patients or the public were not involved in the design, conduct, or reporting of this research.

## Ethical statement

Ethical approval was not applicable for this study as we conducted a scoping review of published studies.

## Results

The search yielded 419 studies published between 2000 and 2025 (Fig 1). Twenty-six studies on integrated care models in African settings were included in this review [29–54], of which 31% (8/26) were observational studies, and 27% (7/26) were randomized studies and registered trials (Table 1). The randomized studies were of moderate quality as per the CASP checklist. Most studies were recent and published between 2020 and 2025 (85%, 22/26). The most common integrated care model was the HIV-Hypertension-Diabetes model (34%, 9/26 studies), followed by the HIV-Hypertension model (26%, 7/26 studies).

The key intervention components of integrated care models in Africa included training and retraining healthcare workers to offer integrated health services, improving data collection and information systems, and developing health education and promotion programs (Table 2). These intervention components were identified as crucial building blocks of an integrated care model, with flexibility to tailor to local contexts. Notably, these components varied in frequency across the various types of integrated care models (Table 2). The least common intervention component was microfinance.

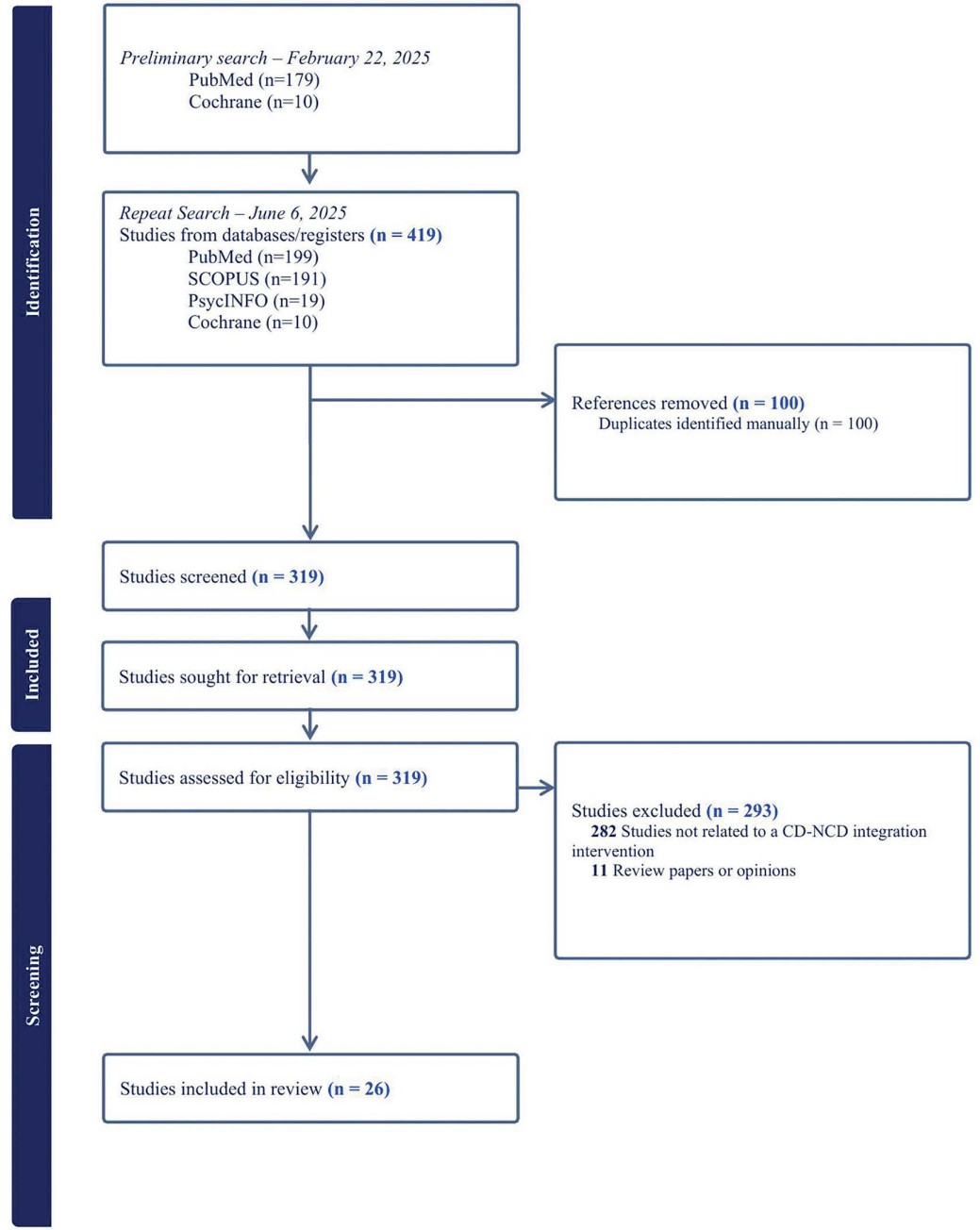

**Fig 1. PRISMA flowchart of included studies.**

Next, we describe real-world implementation of integrated care models under experimental conditions to further illustrate key intervention components and structural and process outcomes compared to standard of care scenarios in various settings. These models illustrate the varied ways integrated care models have been developed and tested in Africa using randomized trials. We focus on these studies to illustrate the state of currently published causal evidence around the implementation of integrated care on health and economic outcomes.

**Table 1. Characteristics of included studies.**

| Characteristic | Number of studies (%) |
|---|---|
| **Region** | |
| Western Africa | 1 (4%) |
| Southern Africa | 6 (23%) |
| Eastern Africa | 19 (73%) |
| North Africa | 0 (0%) |
| **Study design** | |
| Observational study (e.g., cross-sectional, cohort) | 8 (31%) |
| Randomized study (e.g., RCT, cluster RCT) | 7 (27%) |
| Qualitative study (e.g., interviews, focus groups) | 6 (23%) |
| Mixed-methods study | 5 (19%) |
| **Publication year** | |
| 2015 - 2019 | 4 (15%) |
| 2020 - 2025 | 22 (85%) |
| **Number of participants** | |
| < 1,000 participants | 17 (65%) |
| > 1,000 participants | 9 (35%) |
| **Integrated care model type** | |
| HIV, Hypertension and Diabetes | 9 (34%) |
| HIV and Hypertension | 7 (26%) |
| HIV and Cancer | 2 (8%) |
| HIV and Mental health condition | 2 (8%) |
| Diabetes and Tuberculosis | 1 (4%) |
| HIV and NCDs | 1 (4%) |
| HIV, Tuberculosis and Hypertension | 1 (4%) |
| HIV, Tuberculosis, Hypertension, Diabetes and Other NCDs | 1 (4%) |
| HIV, Tuberculosis, Other NCDs and Malnutrition and antenatal care | 1 (4%) |
| HIV, Hypertension, Diabetes and Cancer and Other NCDs | 1 (4%) |

**Table 2. Intervention components appearing in the included studies.**

| Intervention Component | Count and % of Total Appearances |
|---|---|
| Training healthcare workers and capacity building | 23 (88%) |
| Improved data collection and information systems | 21 (81%) |
| Health education and promotion | 19 (72%) |
| Single point of care/ "one-stop shop" | 16 (62%) |
| Availability of medical equipment, diagnostics and testing | 16 (62%) |
| Patient self-management and coaching | 13 (50%) |
| Medication adherence and monitoring | 13 (50%) |
| Community-based programs | 9 (35%) |
| Task-shifting/sharing | 8 (31%) |
| Strong linkage and referral system | 5 (19%) |
| Microfinance | 2 (8%) |

## Experimental models of integrated care

INTE-Africa is a pragmatic cluster-randomized trial in Uganda and Tanzania, where participants were randomized to either standard care (separate services for HIV, diabetes, and hypertension) or integrated care for HIV, diabetes, and hypertension [29]. The authors found that integration of NCD health services into HIV care did not adversely affect HIV outcomes. Additionally, they found that integrated care has a retention rate similar to that of standard care (over 89% in both study groups), but it incurs lower facility costs per patient with multiple conditions compared to the standard care alternative [29]. In their secondary outcomes, blood pressure and glucose indicators improved in the integrated care group compared to standard care, but these findings were only statistically significant for changes in fasting blood glucose from baseline [29]. Identified components of the integrated care alternative included a single set of healthcare workers, the same pharmacy, similarly constructed medical records, and the same laboratory service for all patients receiving integrated care [29]. The integration intervention was regarded non-inferior to stand-alone treatment of chronic diseases.

The SEARCH Trial is an active cluster-randomized controlled trial of streamlined care in Uganda and Kenya [30]. In 16 intervention communities, patients received streamlined care for NCDs in addition to the standard of care for HIV. The authors reported improved blood pressure control among patients referred to the integrated chronic disease care clinic [55]. A key intervention component of their intervention was a nurse-driven triage and care system that aimed to reduce waiting times and offered identical processes of care for HIV, diabetes, and hypertension services [30]. Process outcomes of their multi-disease chronic model included lower costs of care due to integrated services, and more convenient health service-seeking due to their "one-stop shop" approach [30,55]. Another key outcome included reduced stigma associated with integrated care clinics, which the authors attributed to the relabeling of clinics as "medical" clinics instead of "HIV" clinics [30].

The Harambee Trial is a cluster randomized controlled trial of integrated community-based care incorporated into a microfinance intervention in Kenya [31]. NCD management for people living with HIV was incorporated into existing microfinance group meetings, which addressed the issue of geographical availability and accessibility of health services. The intervention included leveraging monthly group meetings for microfinance groups to provide community-based care, point-of-care laboratory testing and medication distribution. Another key intervention component was establishing peer support to enhance ART adherence [31]. A key outcome reported by the authors included modest economic savings and reduced loan defaults among participants of the integrated care/microfinance model [32].

The TASSH trial is an active cluster randomized controlled trial integrating evidence-informed Task-Strengthening Strategy for Hypertension Control (TASSH) into HIV care in Nigeria [33,56]. The intervention is designed to free up physicians' time through task-sharing so that they can focus on more advanced and complex patients, while nurses and community health workers are equipped to handle stabilized patients [56]. One of their key intervention components is practice facilitation, which leverages external expertise to enhance practice redesign to enable more task-shifting/sharing for integrated care. Another key component is equipping health workers with the knowledge and skills to deliver lifestyle counselling for participants identified as at high risk for cardiovascular disease [56].

In South Africa, a cluster randomized trial integrated mental health services with HIV services through collaborative care and task-sharing [34]. Individuals on antiretroviral treatment (ART) were administered the Patient Health Questionnaire-9 (PHQ-9), and referred to depression counselling and treatment if necessary. Key intervention components included training on mental health disorders, leveraging nurses for task-sharing, and increasing communication and collaboration between healthcare workers. The authors reported no effect on depressive symptoms, which they attributed to low referrals to counseling services within the highly pragmatic setting of the study [34].

Lastly, a stepped-wedge cluster randomized trial in Malawi utilized a household-based community health worker model that integrated HIV, NCD, TB and family planning and antenatal care [35]. The intervention expanded an existing HIV-TB program to offer more comprehensive care in a low-resource setting by leveraging community health workers and their networks. One of the intervention design elements included instituting a programmatic shift from a single disease to a

multi-disease focus for community health workers in the region. This included a 5-day foundational training for community health workers to acquaint them with their broadened scope and new tasks [35]. The authors reported a 20% decrease in the number of patients defaulting on NCD treatment [35].

Next, we describe the barriers and facilitators of integrated care models from both experimental and observational studies (Table 3).

## Facilitators and barriers of integrated care

Developing guidelines for health service integration also emerged as a key facilitator of enhanced integration, particularly for healthcare workers who previously focused on single disease areas, such as HIV. Having care guidelines and protocols appeared as key intervention component in 81% (21/26) of included studies (Table 3). A study by Badacho and colleagues reported ten constructs that were found to influence the successful integration of health services in Ethiopia, which included the availability of NCD guidelines [36]. Additionally, a pre-post study by Muddu et al. showed that the WHO-HEARTS intervention helped facilitate an effective HIV-hypertension integration model and argued that adopting the WHO-HEARTS protocol into national guidelines could promote the sustainability of HIV-hypertension integrated models [37]. Similarly, Shayo et al. conducted a qualitative process evaluation of integration in Tanzania and found that developing guidelines on service integration for practitioners was a key facilitator for the sustainability and scaling up of integrated care [38].

Stakeholder engagement appeared as a key facilitator of integrated care in 62% of the studies in the review. Ensuring buy-in and support from key stakeholders, such as healthcare workers, hospital support staff, community members, policymakers, and other key stakeholders was noted as crucial for enabling effective integration of health services. For instance, Kivuyo et al. described conducting extensive community engagement before, during, and after the trial of their HIV-diabetes-hypertension integrated care model, which they noted as a key contributor to high retention rates in the program [29].

Task-shifting and task-sharing was noted as another key facilitator of integrated care in 50% of the studies. By shifting tasks from physician-level healthcare workers to nurses and community health workers, healthcare worker time is spread out and more patients can be seen. Harrison et al. describe a conceptual model of delivering integrated care, particularly for HIV and diabetes, and specifically highlighted the critical role of task-shifting as a facilitator in integrated care [57]. Additionally, Kathree and colleagues describe a comparison group cohort study on the integration of depression and HIV care in South Africa, where they also identified task-sharing and collaborative care as an important facilitator of integrated care models in low-resource contexts [39].

Notably, community health workers (CHWs) were seen as a valuable resource for facilitating integrated care, particularly in low-resourced settings (n = 10, 38%) [35]. A study of community health workers' by Rachlis et al. noted that CHWs could encourage health-seeking behaviors and testing for NCDs and provide accessible education at the community level [40]. Another study by Babalola et al. provided additional evidence of the potential benefits of training CHWs to provide

**Table 3. Facilitators and Barriers of Integrated Care models and CFIR level, shown as counts and percentage of total appearances.**

|  | Innovation | Individual | Inner Setting | Outer Setting | Implementation |
|---|---|---|---|---|---|
| **Facilitator** | • Community health workers (n = 10, 38%)<br>• Technology (n = 8, 31%)<br>• Collaborative care (n = 7, 27%) |  | • Care guidelines (n = 21, 81%)<br>• Task-shifting (n = 13, 50%) | • Financing (n = 7, 27%) | • Stakeholder engagement (n = 16, 62%) |
| **Barrier** | • Limited equipment, medicines, and diagnostics (n = 17, 65%)<br>• Costs of care (n = 13, 50%) | • Limited health and functional knowledge (n = 17, 65%) | • Limited staff and key healthcare personnel (n = 20, 77%) | • Supply Chain Issues (n = 9, 35%) |  |

comprehensive care across a range of diseases, which demonstrates the potential for expanding integrated care outside typical healthcare settings [41].

There were also several barriers to the integration of CD-NCD health services noted in the literature. Limited staff and key healthcare personnel emerged as a barrier to integration in 77% of studies. One study highlighted shortages of human resources, such as specifically trained practitioners, as key barriers to the integration of TB and diabetes services in Malawi [42]. In addition to low numbers of healthcare workers, there were noted fears of increased workload among providers [43]. Particularly, there were concerns that integrated services without sufficient restructuring of workflows would overburden existing healthcare workers [43]. Weak patient tracking mechanisms were also noted as a key challenge, which is related to the lack of human resources to coordinate care and enhance communication between healthcare personnel and information systems [33].

Additionally, limited knowledge, particularly among healthcare workers, emerged as a key barrier to integration in 65% of studies. A cross-sectional study of healthcare workers in Botswana reported gaps in knowledge of NCDs among healthcare workers attending to HIV patients, which necessitated more training on NCD services to enhance effective integration [44]. Another study noted that expanded training and capacity building were necessary for the sustainability of integrated care models, particularly for NCD care provision [45].

Limited medicines and medical supplies were also a notable barrier to effective health service integration (65% of studies). Haruna and colleagues conducted a qualitative study of the integration of HIV-NCD care in Tanzania and found that limited and inconsistent supplies of medical equipment were a key barrier to integrated healthcare [46]. Gala et al. also identified insufficient resources as a key barrier to integration, such as a lack of medicines to treat identified NCD patients [47]. Another process evaluation of an integrated care model for HIV, hypertension, and diabetes in Uganda also highlighted that inadequate NCD drug supply chains were a key barrier to integration [58]. The cost of material resources was also important. Particularly, Low and colleagues identified high costs of certain treatments –to the patient— as a barrier to integration, specifically for cancer treatment [48].

Next, we describe the use of digital technology in integrated care models.

## Digital technology

A few studies mentioned the use of digital technology to facilitate the integration of health services. For instance, the SEARCH study utilized mobile phone calls to deliver appointment reminders and offer easy scheduling for clinic visits, which they noted as a key component of the integrated care intervention [30]. The Harambee Trial leveraged an electronic medical record system and mobile tablets with cloud-based data capture to track participants' health status and intervention outcomes [31]. The TASSH intervention in Nigeria leveraged social media channels (WhatsApp) to enhance healthcare workers' engagement with the intervention and encourage shared learning through an online community learning environment [56]. Notwithstanding these examples, only 10 studies (38%) described the use of digital technology in their interventions, demonstrating an area for improvement as the use of integrated care models grows.

## Discussion

As the co-occurrence of two or more health conditions becomes more common in global settings, particularly in Africa, where there is a disproportionate burden of health and life loss due to both communicable and non-communicable diseases, there is a growing need for innovative healthcare delivery pathways that recognize the increasing complexity of patients and communities [13]. Integrated care models have been proposed as a potential health policy solution to address the burden of multimorbidity, and the fragmented nature of healthcare delivery in healthcare systems in Africa and other settings [3]. While there are numerous conceptualizations of integrated care [19], we focused on models that combined or consolidated health services for both communicable and non-communicable diseases. We identified different types of integrated care models that have been implemented in varying contexts across the African continent,

demonstrating general feasibility and acceptability [23]. Key intervention components included training healthcare workers on integration and improving data collection and information systems to facilitate collaboration between healthcare workers. Having care guidelines, particularly for NCD integration, emerged as a key intervention facilitator for enhancing the success of integrated care models.

Our findings show that the integration of care for communicable and non-communicable diseases may deliver several benefits, such as improved blood pressure control [55]. Importantly, a randomized trial showed that the integration of diabetes and hypertension services into HIV care did not negatively affect HIV outcomes [29], which demonstrates that the successes gained in HIV care can be sustained even with expansion of care into other disease areas. However, another study found that the integration of mental health services into HIV care had no effect on depression outcomes, which was attributed to low referrals and linkages to counselling services [34]. Overall, as corroborated by a review by Chireshe and colleagues, empirical evidence on integrated care is varied and limited, which warrants more investigation in this area [18]. Nonetheless, we found qualitative evidence that shows general support and positive attitudes towards integrated care across the African region. For instance, in a Survey of 195 cancer providers across 4 countries (Malawi, Zimbabwe, Uganda, and South Africa), over 80% of providers were in support of HIV services and information being integrated into cancer care [49].

However, this review also identified several gaps in the literature related to integrated care models. First, the cost-effectiveness of integrated care models is still uncertain [3]. Nugent and colleagues note that adding NCD screening to HIV clinics may cost more in some instances, but may be cost-effective in the long-run [59]. However, there is a dearth of empirical evidence on the cost-effectiveness of integrated care models, particularly in Africa [11,59]. Moreover, the economic implications on different stakeholders is still unknown, i.e., the cost-to-benefit ratio for patients, providers, governments, etc. As such, more economic evaluations of integrated care are needed to elucidate the trade-offs of integrated healthcare delivery pathways [60]. Importantly, equity considerations in cost-effectiveness analyses of integrated care models are crucial, as there is evidence that multimorbidity may follow social gradients, i.e., demonstrate more prevalence in lower socio-economic groups [13]. Therefore, integrated care efforts to address multimorbidity will require equity-informative evaluations.

Moreover, the attempts to integrate services for communicable and non-communicable diseases have not been without challenges [36,50,61–63]. The literature frequently notes the lack of human resources and medication shortages as barriers to integrated care. Notably, strengthening the structural and operational capacity of healthcare centers is crucial, particularly through recruiting NCD-trained healthcare workers and strengthening supply chains for medicines, diagnostics, and equipment [51]. Importantly, structural and operational capacity is important for retaining patients in care. Muddu et al. reported on a retrospective cohort study of people living with HIV in Uganda who also received hypertension treatment, and found that only about half were retained in care, noting that integration of health services is not enough without strengthening the capacity of healthcare centers to retain patients in care [52,53].

Notably, most studies in this review focused heavily on hypertension, diabetes, and HIV, and less on other NCDs and infectious diseases. This is an opportunity for investment as the prevalence of NCDs and multimorbidity grows in African region and globally, necessitating wider-scoped care delivery models. For instance, the literature has shown that women with multimorbidity, i.e., HIV and cancer have worse breast cancer survival compared to women without multimorbidity, which necessitates healthcare delivery models that are responsive to the rise of NCDs, particularly cancer, in Africa [64]. Another study of TB-diabetic patients in Tanzania reported on the urgent need for integrated care models for patients suffering from TB and diabetes comorbidity, which is evidence of the need to expand on current HIV-based models to include other health concerns in the population [65]. Importantly, health system innovation that caters to the needs of the whole population, while remaining proportional to the urgent needs of identified target populations, is fundamental. Particularly, it is crucial to include sexual and reproductive health programs in integrated care models. For instance, only one study in our review described integrated services for family planning and antenatal care into HIV-NCD care [35], demonstrating a gap in this area.

Moreover, several scholars have emphasized the need for a more nuanced conceptualization of "integration," particularly noting that the binary contrast of horizontal and vertical programs is a limited view [19]. For instance, a conceptual framework for integration by Atun and colleagues proposes several facets that can be used to deconstruct integration according to functions served in the health system [19]. Under this framework, the interventions described by our review fall within the 'health service delivery' function of the health system, specifically at the initial stage of combining services that are perceived as more efficiently addressed together, such as cardiometabolic issues (diabetes and hypertension, for instance). However, there are several steps beyond combination of services that are necessary for true whole health system transformation, i.e., adaptation, evaluation, scale-up [54], etc. Notably, most of the studies noted in our review were most often small-scale interventions, some experimental, and limited to one or two health facilities at most, demonstrating that there is still a long road ahead to achieve health system integration and transformation.

This review also presents results on the potential for digital technology to enhance integration in Africa. A few studies leveraged mobile phone technology and electronic health records to facilitate integrated care, however, there is a need for more technological advancement in this area. Importantly, the growth of artificial intelligence (AI) may present opportunities and challenges for digital health approaches in integrated care [66]. Particularly, AI may be used to identify patients at-risk for multimorbidity. For instance, Isangula et al. described the use of AI and machine learning to detect coughs, an intervention which could be leveraged in integrated care settings to identify patients developing additional morbidities [67]. As the use of AI grows, there may be additional opportunities to facilitate digital health in integrated care models, however, this must be balanced with additional considerations to ensure data protection and accuracy [66].

## Policy implications

This review has several implications for policy and intervention development in Africa, as well as globally. First, integrated care may be a potential antidote to the single-disease approach of providing care separately for communicable and non-communicable diseases, which has been the norm in many African healthcare systems [3]. Specifically, the World Health Organization has called for a move towards integrated people-centered care, and adopted a framework to advance this area in 2016 [5]. This review provides practical features of the key elements of integrated care models, which can be used to inform development and implementation of integrated care.

Moreover, as donor funding in global health diminishes, integrated care may offer a way to optimize available health resources to ensure continued access to care in Africa. The experimental models highlighted in this review, mostly built on HIV chronic care provision, can serve as blueprints for future models that optimize available health resources to innovate and expand health services to other diseases. For instance, greater investment in primary care structures can allow primary care practitioners to offer more needed services at the first line of care [18]. In the global arena, some of the integration models identified in this review may inform the development of innovative models in underserved areas in high-income countries, through the function of "reverse innovation." That is, the finding of innovative solutions to multimorbidity management in low-resourced settings, such as the integrated care models referenced in this review, can be used to inspire innovative delivery models in higher-resourced settings.

## Strengths and limitations

This review has notable strengths. First, the search was widely inclusive in time frame, as all studies published between 2000 and 2025 (up to June) were considered. Additionally, each included study was double-coded and quality-assessed to improve the accuracy of data extraction and reduce bias. Importantly, the review considered a broad range of both communicable and non-communicable diseases, whereas prior reviews have mostly focused only on HIV/AIDS and a limited set of NCDs [18,23,68,69]. Additionally, this review considered integrated care for mental health conditions, which still receive fragmented attention across various health settings. Although our review was explicitly designed to examine integration of communicable and non-communicable disease services, the near-exclusive focus of identified models on

HIV care –despite well-documented integration efforts for tuberculosis, malaria, and other infectious diseases– reveals a substantive gap in the empirical, peer-reviewed literature. We believe this to be an important finding of this review.

The review also had some limitations that may be areas of future research. First, we limited our review to peer-reviewed papers written in the English language and used mostly English-based databases. Although our language restriction to English did not exclude any peer-reviewed papers in the databases we searched, we did not search any literature in other languages. Future reviews may benefit from searching non-English literature, as well as grey literature. Secondly, the review excluded integrated care models that focused only on infectious diseases and those focused only on NCDs. Although the objective of the review was to elucidate CD-NCD integration, there are numerous other kinds of integrated care models that we may have missed due to the search criteria. Additionally, this review did not comprehensively analyze intervention outcomes, as we sought rather to investigate intervention components to inform future implementation efforts. Moreover, the study of intervention components, facilitators, and barriers using frequency counts does not indicate the strength of the evidence. As such, there is still a need for a more in-depth study of the components of integrated care interventions, including conceptual mapping through logic models and theory of change models, to identify how and why integrated care may or may not deliver better outcomes for people with multimorbidity.

Throughout the research and development pipeline, there is a need for a more robust and nuanced view of "integration" to guide broader health system innovation and move us towards more holistic, patient-centered care at all levels. Particularly, researchers, policymakers and practitioners ought to consider integrated models that move beyond simply adding more health services but rather transform healthcare delivery to better align with community needs. Funders of research and programs play a key role in this endeavor through working to reduce power asymmetries and pursuing what others have called *authentic allyship* [70]. Indeed, to enhance health system transformation in nations across the continent, true partnerships focused on demonstrated need, rigorous science, and community-led wisdom will be critical.

## Conclusion

As multimorbidity becomes increasingly prevalent, particularly in the African region, there is a need for innovative healthcare delivery models that address the complexities of managing both communicable and non-communicable diseases. This review identifies different types of integrated care models that have been implemented in various contexts in the African region, with notable benefits such as improved health outcomes and lower facility costs per patient. However, challenges such as limited human and material resources persist. Moreover, most studies primarily focused on hypertension, diabetes, and HIV, demonstrating an opportunity to expand integrated care to other conditions such as cancer. Additionally, while some studies have begun to explore the use of digital technology to enhance integrated care, further research is needed to improve these interventions in both African and global contexts.

## Supporting information

**S1 Appendix – Search Strategy.**
(DOCX)

**S2 Appendix – Use of Artificial Intelligence.**
(DOCX)

**S1 Checklist, PRISMA Checklist [25].**
(PDF)

## Author contributions

**Conceptualization:** Takhona G. Hlatshwako.

**Data curation:** Takhona G. Hlatshwako, Oluwabusolami Ale, John Chapola, Lirui Jiao, Shuyi Song, Audrey Yao, Tara Templin.

**Formal analysis:** Takhona G. Hlatshwako.

**Funding acquisition:** Takhona G. Hlatshwako.

**Investigation:** Takhona G. Hlatshwako, Oluwabusolami Ale, John Chapola, Lirui Jiao, Shuyi Song, Audrey Yao, Leah Frerichs, Tara Templin.

**Methodology:** Takhona G. Hlatshwako, Leah Frerichs, Tara Templin.

**Project administration:** Takhona G. Hlatshwako.

**Supervision:** Leah Frerichs, Tara Templin.

**Writing – original draft:** Takhona G. Hlatshwako.

**Writing – review & editing:** Takhona G. Hlatshwako, Oluwabusolami Ale, John Chapola, Lirui Jiao, Shuyi Song, Audrey Yao, Leah Frerichs, Tara Templin.

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
