## [Decision Letter · Decision Letter 0]

30 Nov 2025

PGPH-D-25-02635

The integration of communicable and non-communicable disease (CD-NCD) health services in Africa: A Scoping Review

Dear Dr. Hlatshwako,

Thank you for submitting your manuscript to PLOS Global Public Health. After careful consideration, we feel that it has merit but does not fully meet PLOS Global Public Health’s publication criteria as it currently stands. Therefore, we invite you to submit a revised version of the manuscript that addresses the points raised during the review process.

The manuscript has been evaluated by one reviewer and their comments are available below.

They raised a number of concerns on the critical analysis within the manuscript and provision of further information from the literature search. Could you please carefully revise the manuscript to address all comments raised?

Please note that we have only been able to secure a single reviewer to assess your manuscript. We are issuing a decision on your manuscript at this point to prevent further delays in the evaluation of your manuscript. Please be aware that the editor who handles your revised manuscript might find it necessary to invite additional reviewers to assess this work once the revised manuscript is submitted. However, we will aim to proceed on the basis of this single review if possible.

We look forward to receiving your revised manuscript.

Kind regards,

Jen Edwards

Staff Editor

Journal Requirements:

Additional Editor Comments (if provided):

Reviewers' comments:

Reviewer's Responses to Questions

**Comments to the Author**

1. Does this manuscript meet PLOS Global Public Health’s publication criteria ? Is the manuscript technically sound, and do the data support the conclusions? The manuscript must describe methodologically and ethically rigorous research with conclusions that are appropriately drawn based on the data presented.

Reviewer #1: Yes

2. Has the statistical analysis been performed appropriately and rigorously?

Reviewer #1: N/A

3. Have the authors made all data underlying the findings in their manuscript fully available (please refer to the Data Availability Statement at the start of the manuscript PDF file)?

Reviewer #1: Yes

4. Is the manuscript presented in an intelligible fashion and written in standard English?

Reviewer #1: Yes

Reviewer #1: This is an interesting and useful paper that summarizes the state of the evidence on integration of communicable and noncommunicable diseases within healthcare systems in African countries.

Overall, the methodology is sound, and the results are in line with what I would expect from such a review. However, I have two substantive critiques and a few minor points.

The first critique is that I think the paper falls short by not offering a more critical assessment of the construct of "integration" and how it is used in the studies cited in this review.

What do I mean?

There are at least two different ways one could conceptualize integration. One is to look holistically at different health system functions and the pros and cons of using vertical/targeted approaches to deliver certain services. A couple of articles by Atun, Adeyi, and colleagues from 2010 in Health Policy and Planning summarize this way of thinking. The other is to focus on "adding on" new services to a set of existing services, often for a given (priority) patient population. Much of the literature around HIV has gone this direction - for example, see a July 2018 supplement in AIDS. This scoping review aligns implicitly with the latter, and as expected, the literature uncovered in this scoping review almost exclusively focuses on adding services for PLHIV with non-HIV health needs.

While there is nothing wrong with improving primary care for PLHIV, this conceptualization of integration falls massively short as a way of guiding health system development. PLHIV are indeed a priority population group, but they only represent 2.5% of the population, and they have been fortunate to have a lot of resources put into developing their own healthcare system that largely operates in parallel to the general healthcare system. What about the other 97.5% of the population? The reality is that chronic disease services are not fragmented -- they're simply unavailable. I am not sure that doing a few in-service trainings or digital health pilots by implementation scientists can fix the situation for this 97.5% of the population. What is needed, I think, is an overhaul of pre-service training of PHC nurses so that they have the skills needed to do basic chronic disease screening and treatment. And over the long term, national governments need to invest in the family medicine PHC model, where one primary care provider (doctor or advanced-practice provider) is able to do first-line evaluation and management of pretty much any health concern a patient comes in with. That doesn't obviate the need for vertical/specialized workforce and institutions, eg, dedicated TB clinics or cancer hospitals. But I think the evidence is clear that it's the most efficient and sustainable way to build an integrated healthcare system over the long term.

Put another way, we're having this conversation about "integration" in African countries because we're still dealing with the legacy of colonialism and neo-colonial ideas about how a health system should be designed (and which patients should receive priority for services). I think we need to be pushing researchers to think about integration in a much broader way than simply teaching ART nurses to screen for hypertension.

I don't expect the authors to solve this problem, or to change the substance of their review. But I think we need to be slowly shifting the discourse around integration to focus on patient-centered care for everyone, rather than making tweaks to donor initiatives. In my view, there is a bit of groupthink happening among HIV-adjacent researchers, many of whom (at least till recently) have been funded predominantly by American institutions like the NIH and PEPFAR that are very disease-focused instead of health system-focused. When they talk about integration, they are talking about adding this or that service within ART clinics, without asking bigger questions about how to build a people-centered health system.

For this scoping review, I would hope for a few additional sentences here and there in the results and discussion sections that underscore these following points (assuming the authors agree):

(1) there are different ways of thinking about integration (see my first point above);

(2) the "model types" that the authors have uncovered (Table 1) are mostly focused on adding services one- or two-at-a-time into HIV care, and these still need to be integrated amongst themselves (!); and

(3) this heavy focus on incremental improvements in care for PLHIV has led to major blind spots in the literature about how to develop PHC systems that provide essential services for the entire population.

The second critique is that I'm not sure if a disease or risk factor constitutes a "model" per se (as per Table 1). I think models of care are better defined around human resources (eg, competencies/scope of practice) and the geospatial arrangement of different components of care (eg, dispensaries vs. health centers vs. hospitals). The authors might wish to have a look at a scoping review done recently on integrated care models for NCDs and mental health conditions (https://pubmed.ncbi.nlm.nih.gov/36717832/) and consider doing a bit of additional classification/cross-tabulation of the models of care described in the literature they identified.

Finally, I offer a few smaller points for consideration:

There is no explicit mention of a grey literature search. I could imagine some reports on NGO websites or other parts of the internet on real-world care delivery that were never written up as peer-reviewed papers.

The English language restriction should be defended more than it is currently. What proportion of the literature is potentially being missed by not considering studies (including, importantly, in the grey literature) from francophone, arabophone, and lusophone countries?

The paper mentions "African region" but does not define the countries referred to. Do they mean sub-Saharan Africa, WHO-AFRO, or something else? The appendix search strategy implies AFRO, but it would be good to say this in the main text.

Finally, I note that the barriers (and to a lesser extent, enablers) listed in these studies (Table 3) are pretty generic and could describe health system gaps in general, not just CD-NCD care. There's been a gazillion studies on barriers for various diseases, with a lot of overlap. A brief nod to this literature, at least as relates to chronic disease, would be useful.

(what does this mean? ). If published, this will include your full peer review and any attached files.). If published, this will include your full peer review and any attached files.

**Do you want your identity to be public for this peer review?** For information about this choice, including consent withdrawal, please see our Privacy Policy .

Reviewer #1: **Yes:** David WatkinsDavid Watkins

---

## [Decision Letter · Decision Letter 1]

26 Jan 2026

PGPH-D-25-02635R1

The integration of communicable and non-communicable disease (CD-NCD) health services in Africa: A Scoping Review

Dear Dr. Hlatshwako,

Thank you for submitting your manuscript to PLOS Global Public Health. After careful consideration, we feel that it has merit but does not fully meet PLOS Global Public Health’s publication criteria as it currently stands. Therefore, we invite you to submit a revised version of the manuscript that addresses the points raised during the review process.

While the article explores a pertinent topic, we feel that there is need to re-word some of the content to accurately reflect the results of the review. Please address the additional comments shared below.

We look forward to receiving your revised manuscript.

Kind regards,

Andrew Kazibwe, MBChB, MMED

Academic Editor

Journal Requirements:

1. Please provide a detailed online Financial Disclosure statement. This is published with the article. It must therefore be completed in full sentences and contain the exact wording you wish to be published.

a) State the initials, alongside each funding source, of each author to receive each grant. For example: “This work was supported by the National Institutes of Health (####### to AM; ###### to CJ) and the National Science Foundation (###### to AM).”

For more information, please go to our submission guidelines:

https://journals.plos.org/globalpublichealth/s/submission-guidelines#loc-financial-disclosure-statement

2. Please ensure that the funders and grant numbers match between the Financial Disclosure field and the Funding Information tab in your submission form. Note that the funders must be provided in the same order in both places as well.

3. Please update your online Competing Interests statement. If you have no competing interests to declare, please state: “The authors have declared that no competing interests exist.”

4. We note that your Data Availability Statement is currently as follows: "The data was compiled from published studies. The protocol was registered in OSF (https://doi.org/10.17605/OSF.IO/ZMR5K)."

Please confirm at this time whether or not your submission contains all raw data required to replicate the results of your study. Authors must share the “minimal data set” for their submission. PLOS defines the minimal data set to consist of the data required to replicate all study findings reported in the article, as well as related metadata and methods (https://journals.plos.org/globalpublichealth/s/data-availability#loc-minimal-data-set-definition).

If your submission does not contain these data, please either upload them as Supporting Information files or deposit them to a stable, public repository and provide us with the relevant URLs, DOIs, or accession numbers. For a list of recommended repositories, please see https://journals.plos.org/globalpublichealth/s/recommended-repositories.

5. Please include a separate legend or caption for Figure 1 in your manuscript.

Additional Editor Comments (if provided):

Reviewer 2

This is a timely and exciting article. Below are a few comments for consideration:

(a) Whereas the authors aimed to study communicable and non-communicable disease integration, their search seems to have yielded results pertinent to HIV care only, in the communicable disease domain. It is therefore prudent that the study title and introduction reflect this bias (which is justified by the significant cumulative investment in HIV care systems). Other communicable diseases where integration has been key include viral hepatitis, tuberculosis, malaria and bacterial sexually transmitted infections among others. The fact that these did not feature in the article negates the broad focus on communicable diseases as portrayed in the title and background sections.

(b) There is a discrepancy in the numbers of articles retrieved and included in the protocol referenced, and those reported in the review. These ought to be harmonised, and any discrepancies explained.

(c) What was the basis for labeling the integration models based on disease entities, rather than service locations?

(d) In the methodology section, the authors should clearly describe what data was extracted from each source.

Reviewers' comments:

Reviewer's Responses to Questions

**Comments to the Author**

Reviewer #1: All comments have been addressed

publication criteria ? Is the manuscript technically sound, and do the data support the conclusions? The manuscript must describe methodologically and ethically rigorous research with conclusions that are appropriately drawn based on the data presented.

Reviewer #1: Yes

3. Has the statistical analysis been performed appropriately and rigorously?

Reviewer #1: N/A

4. Have the authors made all data underlying the findings in their manuscript fully available (please refer to the Data Availability Statement at the start of the manuscript PDF file)?

Reviewer #1: Yes

5. Is the manuscript presented in an intelligible fashion and written in standard English?

Reviewer #1: Yes

Reviewer #1: The authors have done a great job responding to my original critiques. I have no further comments.

(what does this mean? ). If published, this will include your full peer review and any attached files.). If published, this will include your full peer review and any attached files.

**Do you want your identity to be public for this peer review?** For information about this choice, including consent withdrawal, please see our Privacy Policy .

Reviewer #1: **Yes:** David WatkinsDavid Watkins

---

## [Editor Report · Decision Letter 2]

6 Feb 2026

PGPH-D-25-02635R2

The integration of communicable and non-communicable disease (CD-NCD) health services in Africa: A Scoping Review

Dear Dr. Hlatshwako,

Thank you for submitting your manuscript to PLOS Global Public Health. After careful consideration, we feel that it has merit but does not fully meet PLOS Global Public Health’s publication criteria as it currently stands. Therefore, we invite you to submit a revised version of the manuscript that addresses the points raised during the review process.

To demonstrate transparency in the conduct of the review, we encourage you to make minor revisions. These are intended to reflect changes to the review methodology.

We look forward to receiving your revised manuscript.

Kind regards,

Andrew Kazibwe, MBChB, MMED

Academic Editor

Journal Requirements:

Additional Editor Comments (if provided):

I thank the authors for addressing my previous comments and for the additional insight on the number of searches conducted. However, this might be perceived as a lack of transparency in the review process. I therefore suggest that the authors revise the PRISMA-ScR diagram to reflect the additional sources that were included in the review after the repeat search conducted in June 2025 and to share a revised review protocol to reflect these changes. This revised protocol should reference the original protocol and replace the original protocol as the guiding document for the review.
---

## [Editor Report · Decision Letter 3]

17 Feb 2026

The integration of communicable and non-communicable disease (CD-NCD) health services in Africa: A Scoping Review

PGPH-D-25-02635R3

Dear Ms. Hlatshwako,

We are pleased to inform you that your manuscript 'The integration of communicable and non-communicable disease (CD-NCD) health services in Africa: A Scoping Review' has been provisionally accepted for publication in PLOS Global Public Health.

Best regards,

Andrew Kazibwe, MBChB, MMED

Academic Editor